# Artificial Neural Networks in MPPT Algorithms for Optimization of Photovoltaic Power Systems: A Review

**DOI:** 10.3390/mi12101260

**Published:** 2021-10-17

**Authors:** César G. Villegas-Mier, Juvenal Rodriguez-Resendiz, José M. Álvarez-Alvarado, Hugo Rodriguez-Resendiz, Ana Marcela Herrera-Navarro, Omar Rodríguez-Abreo

**Affiliations:** 1Facultad de Informatica, Universidad Autónoma de Querétaro, Querétaro 76230, Mexico; cvillegas22@alumnos.uaq.mx (C.G.V.-M.); mherrera@uaq.mx (A.M.H.-N.); 2Facultad de Ingeniería, Universidad Autónoma de Querétaro, Querétaro 76010, Mexico; jmalvarez@uaq.mx (J.M.Á.-A.); hugorore@uaq.mx (H.R.-R.); 3Red de Investigación OAC Optimización, Automatización y Control. El Marques, Querétaro 76240, Mexico; omar.rodriguez@upq.edu.mx

**Keywords:** neural networks, maximum power point tracking, photovoltaic systems, neuro-fuzzy, hybrid algorithms

## Abstract

The use of photovoltaic systems for clean electrical energy has increased. However, due to their low efficiency, researchers have looked for ways to increase their effectiveness and improve their efficiency. The Maximum Power Point Tracking (MPPT) inverters allow us to maximize the extraction of as much energy as possible from PV panels, and they require algorithms to extract the Maximum Power Point (MPP). Several intelligent algorithms show acceptable performance; however, few consider using Artificial Neural Networks (ANN). These have the advantage of giving a fast and accurate tracking of the MPP. The controller effectiveness depends on the algorithm used in the hidden layer and how well the neural network has been trained. Articles over the last six years were studied. A review of different papers, reports, and other documents using ANN for MPPT control is presented. The algorithms are based on ANN or in a hybrid combination with FL or a metaheuristic algorithm. ANN MPPT algorithms deliver an average performance of 98% in uniform conditions, exhibit a faster convergence speed, and have fewer oscillations around the MPP, according to this research.

## 1. Introduction

The consumption of electrical energy has intensified significantly due to the massive increase in the world population. The use and application of renewable energy solve one of the concerns of the global community. Because the use of fossil fuels is no longer sufficient [1], of all renewable energy sources, solar energy has the best future [2,3]. Photovoltaic (PV), in particular, generates light and electricity from the sun’s energy [4], is expanding rapidly due to supportive policies by governments and, recently, the drastic reduction in its cost [5]. It is also a reliable and commercially available technology with significant long-term growth potential in almost all world regions. The Organisation for Economic Co-operation and Development (OECD) estimates that by 2050, PV systems could provide about 12% of global electricity production and avoid emitting 2.3 Gigatonnes of carbon dioxide into the atmosphere each year [6,7].

However, PV presents notable disadvantages, such as a high initial installation cost and a poor performance producing energy; from 12% up to only 25% of the solar energy is being transformed into electricity [5,8]. Therefore researchers have looked for ways to maximize the total amount of electric energy produced, such as printable thin solar cells [5,9], or using maximum power point tracking (MPPT) devices. The MPPT controllers are a core feature in a PV system, and it maximizes the power output and, as a result, maximizes the overall efficiency of the PV module [4]. An MPPT device consists of two parts, a DC/DC converter and an embedded electronic system with a control algorithm. It is integrated into all PV installations to extract the maximum amount of energy available under all possible environmental and operating conditions and to do so in real time. Control algorithms change in many aspects, such as their ANN architecture, their ease of implementation, the number of sensors required, the range of effectiveness, whether they can follow continuous changes in G and T, the cost, how popular they are, or whether they were implemented in hardware [10]. Various articles and primary papers are analyzed from a taxonomic point of view, and they work as a starting point to further understand the topic of PV control.

The most popular algorithms for MPPT control in PV systems are Perturbation and Observation (P&O) [11], Incremental Conductance (InC) [12] and Constant Voltage (CV) [13]. They have the advantage of being very easy to implement and work very well under uniform radiation and temperature conditions. Nevertheless, as all techniques have disadvantages, they are impossible to adapt to constant changes in the environment and fail to find the Maximum Power Point (MPP) [14]. From these, several improvements are derived to optimize the MPP calculation, such as those using the variable step principle detecting the area under the MPP curve using the derived $\frac{dP}{dV}$ values and a scaling factor. However, failures were also detected in finding the MPP at different radiation levels [15].

Other algorithm improvements use different branches of artificial intelligence, such as those using Support Vector Machines (SVM) [16,17] or Fuzzy Logic (FL) [18,19]. FL algorithms have the advantage of being based on knowledge bases of the PV system and environmental conditions. However, with the disadvantage that their performance is highly dependent on experience in determining rules and membership functions. Other disadvantages described by [18] are the inability to respond quickly at times of changing solar irradiance (G). With low G, the accuracy of the MPPT decreased.

Algorithms for hot spot detection in PV panels and how they affect MPP search and overall system output have also been proposed [20]. Having different PV panels within the array and having one or more present hot spots can reduce system performance by as much as 70%. The articles reviewed in the previous sections do not take this point into account. Refs. [21,22] make a review of classical techniques that already address this issue.

In recent years, metaheuristic algorithms have been considered, primarily designed to solve complex problems with multiple variables and obtain the most optimal possible values [23]. Algorithms such as Ant Colony (ACO) [24,25], Cuckoo Search (CS) [26,27], Firefly Algorithm (FF) [28,29], Whale Optimization (WO) [30,31], Differential Evolution (DE) [32], Gray Wolf (GWO) [33,34] and Particle Swarm (PSO) [35,36]. These were proposed as they can detect the Global Maximum Power Point (GMPP) by combining possible solutions or using random variables. These algorithms apply several particles in each iteration to find the possible solution. A large number of particles may increase the accuracy or speed. Nevertheless, at the same time, it can produce lower energy tracking. In addition, unwanted energy fluctuation is generated due to random variables for the combination and selection of the solution [37].

ANN are used in MPPT controllers to predict the voltage (V) or power (P) output at any time. The calculated value is compared with the instantaneous values obtained to determine the load cycle. Independent variables such as temperature (T) and G will be the input variables of the first layer of the network. Furthermore, other variables such as V and I of the panel can be included as input. The hidden layers will be in charge of processing them. The final performance will depend on the number of neurons in the hidden layers, the chosen activation function, and the desired training algorithm. To further increase the accuracy of the ANN, a good amount of data should be gathered for processing [4].

One of the research challenges faced by the computer scientist or computer systems student when researching control algorithms using ANN is the type of focus of the papers (articles, books, theses) that are written. A mechatronic/electronic approach may focus only on materials, actuators, or sensors. On the other hand, an approach that talks about renewable energies may focus on power generation or a new material that transforms energy, that one whose objective is to show the algorithms and architectures used to achieve the optimization of an intelligent controller and that talks about Machine Learning or Deep Learning (DL) [38]. Another factor affecting this type of work is reproducibility [39]. In 2016, a survey conducted by “Nature” demonstrated that 70% of researchers have tried and failed to reproduce another scientist experiments, and more than half have been unable to replicate their experiments [40]. Researchers are shifting paradigms such as explainability, which is the degree to which a model predictions can be explained in straightforward human terms, toward models that only evaluate performance, in general, using metrics. Having the source code or at least a flowchart available, as well as the hyperparameters and the necessary training data, with the research work, helps validate the algorithms presented. However, this is not a requirement for publishing or presenting at conferences.

The objective of this article is to perform an analysis of the state-of-the-art techniques applied in MPPT devices based on ANN or ANN-hybrid (metauristic) models. The review consists of presenting the trend of the techniques through a search for relevant works of the last 7 years and highlighting the performance of the algorithms from the computational systems approach. A table with the network architecture accompanied by other relevant data such as the percentage efficiency of the MPPT controller, the calculated error, and the type of network training is presented. A summary of each of the works found, and present a block diagram of the algorithm implementation is shown. Finally, this review identifies areas of opportunity to face challenges for future study of intelligent algorithms and MPPT neuro control in PV systems.

This article also attempts to answer the following key questions:What are the most commonly used types of ANN architecture to calculate the MPPT of a PV system?Are existing ANN algorithms suitable for calculating the MPP of a PV system?

The article is structured as follows: Section 1, an introduction to the topic, its importance, and background, in addition to presenting the contribution that this article intends to achieve. Section 2, gives a brief introduction to ANN and some nomenclatures used by PV solar energy systems. Section 3, the summary of the articles whose characteristics describe the MPPT controller that uses ANN. Section 4, the outline of the articles whose characteristics describe MPPT controllers that mix ANN + FL. Section 5, presents the papers whose characteristics describe MPPT controllers that use hybrid techniques (HIS). Section 6 presents the conclusions and discussions about this article, future works, terminologies, and finally, the bibliographical references.

## 2. Theoretical Bases

This section is divided into subsections. Its purpose is to show a brief introduction to the topic of ANN and FL techniques and give a broader context to specific nomenclatures for a better comprehension.

### 2.1. Artificial Neural Networks System

An artificial neural network or ANN is a data processing system inspired by biological neurons. It can be derived from the paper by [41]. They modeled a simple neuron see Figure 1 using electrical circuits in 1943, but the technology was limited at the time. ANN has made significant progress since then. It has an outstanding ability to derive meaning from complex data. It can also detect trends and patterns in too complex data to be classified by either human or computer techniques [42].

A trained ANN can analyze the information that has been provided and provide projections and answers to subsequent problems. ANN takes a different approach to problem-solving than conventional predicting algorithms. They are like the brains of human beings, and they adapt to situations that do not have clear algorithmic solutions and are capable of handling imprecise data. They are used in various applications that have irrelevant data, such as robotics, signal processing, pattern recognition, and financial applications. This advantage allows them to excel in some areas that conventional computers often find difficult [43].

Additionally, ANN are considered viable computational models that can be applied to a variety of complex problems. ANN play a critical role in prediction, classification, complex system prediction and control (CPM), regression problems [44].

### 2.2. ANN Architecture

The type of connections, patterns, or structures that ANN has, is called architecture for the organization of neurons that can be grouped into structures called layers. An ANN is a set of these layers.

Three layers can be distinguished for the basic models: the input layer, the hidden layer or layers, and finally, the output layer. The input layer is where data are received, and these can be sensors that pick up signals from the environment. The output layer is the response to all the synaptic processes within the network, and it can be an effector in the case of a robotic system. The hidden layer is in charge of carrying out the processes (calculations, corrections) representing the environment to be modeled [45].

Thanks to their structure, we can speak of single-layer networks (SLNN), composed of a single layer of neurons, and multilayer networks (MLNN), composed of multiple nodes in multiple layers. MLNN depending on their data flow can be unidirectional (feedforward), where information only goes in one direction, and recurrent networks (RNN) or feedforward, where information can go in any direction, even from the output layer to the input layer [46].

The calculation of the ANN weights is not a simple task. In general, the only solution that exists is trial and error. However, for some particular cases such as the following, there are some approximations. To calculate the total number of weights in a single layer feedforward multilayer ANN, the following definition can be used:

Input x size of the hidden layer + size of the hidden layer x size of the output layer.

There are other methods for each particular architecture, which should be investigated before any implementation [47].

### 2.3. Fuzzy Logic System

Within the branch of fuzzy mathematics, there is a branch called fuzzy logic (FL). The term was introduced by [48] in their fuzzy set theory. Although it has antecedents from the 1920s [49,50] where it was known as “logic of infinite values.”. FL is a form of multivariable logic in which the matching values can be any actual number between 0 and 1. It deals with the concepts of “half-truths” or “partial truths,” where the actual value can be in the range of entirely true or completely false [51].

FL is based on the concept that people make decisions that may be imprecise and that do not possess numerical information. Fuzzy models are a way of representing “vagueness” or imprecise information, hence the term fuzzy. This type of model can recognize and use data and information that has certainty [52]. It has been used in fields such as artificial intelligence and control theory.

### 2.4. Photovoltaic Solar Energy Systems

Temperature (T) or ambient temperature refers to the outside temperature which is the “normal” air temperature. In PVS, it can refer to the temperature at which the PV cell or array can operate. Like any electronic component, the performance of the panel can be affected when it becomes too hot. Therefore, higher or warmer temperatures always mean less power output from the system; this loss can be quantified as “temperature coefficient” and indicated by the PV panel manufacturer and varies from model to model [53].

PV effect (PVE) is the process by which light transforms into electricity. The property of some materials achieves the process of absorbing photons and emitting electrons. When solar radiation is absorbed, its energy is transferred to an electron. With this energy, the electron can leave the atom in which it is located and thus become part of a current in an electric circuit [54].

Solar radiance (SI) measures the amount of solar radiation falling on a given surface, is measured in W/m${}^{2}$. On the other hand, solar irradiance (G) measures the amount of solar energy reaching a surface given period of time and is measured in Wh/m${}^{2}$/day [54].

PV cells (PVC) are electronic devices that convert sunlight into DC, which fluctuate depending on the intensity of sunlight. PVC are made of thin films of semiconductor materials, in which the valence electrons are more bound to the nucleus, but with a small amount of energy, they behave as conductors. For practical use, conversion to specific desired voltages to Alternate Current (AC) is required through inverters [54]. Multiple PVC are usually connected to form modules, which are connected to form arrays. The arrays are connected to the inverter, which is the one that produces power at the desired voltage [55].

The PV Inverter (PVI) is the core of the PV, and it belongs to a large group of static converters. It is responsible for transforming the DC direct current, which comes from the PV panel, into AC alternating current compatible with the load requirements, which is used by batteries or electronic devices more known [55]. PVI is generally divided into two types: stand-alone (SAPV) inverters in Figure 2, which are not in connection to the power grid, and (GCPV) inverters that are connected to the public grid. However, the difference is no longer evident presently because modern systems can operate in both ways in the now called hybrid inverters.

An I-V curve characterizes each PV panel or module, see Figure 3 where the maximum power conditions (I${}_{mp}$, V${}_{mp}$) are explained. This curve, which is a standard, is available in the datasheet of each module or PV panel. It is calculated according to the standardized STC test, equal to the following conditions: (1000 W/m${}^{2}$, 25 °C, IAM 1.5) [56].

## 3. Artificial Neural Networks for MPPT Control in PV Systems

To make this review possible, we used academic search engines such as Institute of Electrical and Electronics Engineers (IEEE), Scopus, MDPI, Semantic Scholar, conventional search engines such as Google Scholar or Microsoft Academic, or novel engines such as connected papers, in addition to institutional repositories such as the one from Autonomous University of Queretaro UAQ, National Autonomous University of Mexico UNAM and others, to perform an exhaustive search and analysis of scientific journal articles, conferences, books, Ph.D., MD and bachelor thesis, on MPPT algorithms that make use of ANN and hybrid techniques, to provide the most representative articles in this area. The results were filtered according to the following criteria:Algorithms using ANN.Algorithms using ANN + FL.Hybrid algorithms (ANN plus metahuristic or optimization algoritms).

ANN MPPT control has been widely researched in PV systems. ANN has the ability to perform MPPT under uniform and variable atmospheric conditions. A general diagram of its operation is shown in Figure 4. Additional table with selected characteristics is shown in Table 1.

In [57] the authors developed a neural network-based MPPT algorithm with the backpropagation technique for PV systems with Boost and Cuk converters. The controller showed good performance, very accurate tracking, small oscillation in very changing weather conditions for the two types of converters. The model was verified using Matlab/Simulink.

The work of [58] presents an MPPT algorithm using neural networks with feed-forward multilayer architecture for application to PV panels installed on the roof of a car, where shading changes occur very fast. The neural network is used to automatically detect the global MPP using pre-selected data from the PV system. This algorithm only uses the I and G variables. As more is data collected, the better the response of the ANN to extract the most energy and improve the prediction accuracy. This network is evaluated with 1000 radiation and temperature samples to simulate changing conditions. The algorithm fails to achieve the global MPP when there are intense changes in weather conditions and performs sub-optimally.

The authors of [59] developed an ANN algorithm for MPPT control in PV systems; two approaches were tested: 1.-optimizing the synaptic weights data structure 2.-multithreaded implementation. It was verified that while a one-dimensional matrix for the synaptic weights significantly improves performance, the same cannot be said for the multithreaded application. In fact, according to the measurements, it was deduced that the thread initialization times are more significant than the benefit of parallelizing the computation. The goodness of fit of the algorithm was verified through simulation-based operational data of a solar panel with 700 irradiance and temperature samples. The simulation results converged with the results of the Matlab program, and the code was adapted to a Launchpad Stellaris device. Measurements showed an execution speed of 60 microseconds with the clock frequency set to 80 MHz. The source code for the project is available at [68].

Research from [60] presents an MPPT algorithm that makes use of resilient networks with backpropagation (Rprop-NN) and supervision to limit the short-circuit I (Isc). This method predicts the MPP instantaneously by measuring T and G, which allows it to move to its optimal operating point without any oscillations. The simulation was carried out in PSCAD/EMTDC software and tested against the P&O and InC algorithms. Under partial shading conditions, the algorithm performed well.

Ref. [61] the authors propose the use of recurrent ANN with exogenous inputs (NARX) for the MPPT control of a PV system. Since the actual output of this type of network is found during the training phase, a parallel/serial architecture was developed to obtain it. The results were simulated using Matlab/Simulink with a hidden layer of ten neurons with a tangent-sigmoid transfer function and for the output layer the simple linear function. An implementation of the algorithm on Arduino Mega was also performed. The results show that the ANN algorithm is more efficient than the P&O algorithm to avoid energy losses. The authors mention that the P&O algorithm is good in low radiation conditions because it can charge the battery, so they recommend the hybrid use of both algorithms.

Ref. [62] the authors propose a feedforward ANN algorithm with a hidden layer and three neurons with G, T, and V as input while I is the output. To increase the efficiency of the network, they used data that already showed variations in T and G because ANN are not good at extrapolating properties. Two hundred test data were used to train the ANN, and the results show good performance when calculating the MPP. In addition, the experiment was repeated, but now with 500 pairs of test data, better results were obtained. This shows that the more data to train, the better the response of our system. Although the results are not surprising, the low complexity makes it suitable for implementation on microcontrollers. The pseudocode for the algorithm is provided.

Ref. [63] the authors developed an MPPT algorithm using high-order recursive ANN (RHONN), in addition to including an extended Kalman filter to optimize the necessary weights. The results were simulated in Matlab using the Simscape toolbox. When G is applied to the system, it can be seen how the algorithm starts to converge quickly and find the MPP, reaching an error of 7% of theoretical MPP. A discrete-time sliding mode MPPT control algorithm was proposed to perform a performance comparison, and the algorithm shows a more stable convergence between the mean error and the standard deviation.

In [64], this article introduces an ANN MPPT algorithm with scanning for predicting Global MPP (GMPP). They compared this new approach versus a enhanced P&O with global scanning and a modified InC with fuzzy duty cycle and a change estimator (FLE). The simulation was done in Simulink and showed faster response time, up to three times faster.

Ref. [65] present two algorithms for MPPT control using ANN, the first one using fix-step and the second one using variable-step. The simulations show that using the variable-step algorithm. Good results are obtained even when there are changing atmospheric conditions, and better response times are obtained, compared to the one using fix-step, which obtained very similar results to those obtained using a simple P&O.

In [66] developed a super twisting sliding mode with ANN algorithm for tracking MPP. They mention that since PVS in changing conditions (T/G) behaves as a nonlinear system, a nonlinear controller is required to ensure MPP. An ANN with three hidden layers was used, and reference data from the I-V curves from the panel was used to train. Simulations were done in Matlab/Simulink, and performance of 97% is claimed.

Ref. [67] the author of this thesis develops an ANN algorithm for MPPT control for PV energy on the Arduino Mega platform with wireless communication capabilities using an Xbee module. The chosen architecture was NARX, and it was chosen because its output is not prone to many variations. It has ten neurons in the hidden layer with a sigmoid transfer function. A simulation was also performed in Matlab using the neural toolbox. It was compared against a P&0 algorithm under the same type of conditions. There is no significant difference between the two algorithms, However, a slightly higher amount of power is delivered to the battery by the P&O algorithm than that supplied by the ANN controller. The controller achieves a higher percentage of power delivered to the battery than the neural controller. Training the sensor and neural controller with data around the MPP could generate better results, although it is crucial to emphasize data obtained with a relatively wide margin around the MPP. Full source code can be obteined at [69].

## 4. MPPT Control Using ANN Plus FL

Fuzzy control provides a formal methodology for representing, manipulating and implementing “human logic or thinking” about how to control a system. FL can be used to build controllers for applications that are very difficult to implement in the real world. The focus of fuzzy control is to gain information and understanding of how to obtain the best control process.

The hybrid technique combining LF with ANN, better known as ANFIS Figure 5, is an excellent tool for working with nonlinear systems, where there is linguistic information and data, and where it is not required to know in depth the mathematical functioning of the system [44,70]. ANFIS consist of a set of rules containing the empirical performance of the system, and the training stage of the ANN. Table 2 shows relevant works from the literature about MPPT using ANFIS techniques.

In [71] the author presents an ANFIS-based MPPT controller coupled to a Z-source DC-DC converter, two 9-rules tables were proposed to better determine the stucture of the controller. Real meteorological data was used to train the network without the need of sensors.The simulation done on Matlab shows that the algorithm performs well under variable conditions.

In [72], the research implements an ANFIS-based MPPT controller with a buck DC-DC converter. An Altera EP4CE6E22C8N FPGA was used for this purpose. Thirty-six datasets were collected for the SR-60S 60 W PV module. For the results, we also implemented the CV algorithm and the InC algorithm on the same FPGA. The new algorithm was shown to capture more power in warmer ambient temperatures. To feed the controller, T, G, and I data are required, in addition to constantly needing adjustment with new data.

In [73], the authors compared two MPPT control strategies, one based on FL and with the ANFIS technique. It was found that DC/DC control by Neuro-Fuzzy approach is more reliable than the other approach as it combines FL plus ANN to extract the MPP, taking advantage of the flexibility of the former and the learning capability of the latter.

In [74], the authors developed an MPPT controller based on the ANFIS algorithm. This algorithm only uses one sensor for I, and it uses a DC-AC inverter two-switch flyback model. To obtain the results, a simulation in Matlab/PSIM was performed. This algorithm manages to obtain an estimated error of 0.0034 with 100 epochs. The results show that the algorithm can calculate the MPP under varying atmospheric conditions, achieving an efficiency of 99.89%.

In [75] the researchers designed an ANFIS algorithm for PV systems, which was trained using accurate data obtained from a Sunny Boy controller. Forty-eight thousand five hundred datasets were collected for more than a year, and the data were simulated in Matlab. The input data were G and T. As the results are strongly dependent on the atmospheric conditions, it was necessary to have a significant variation of G and T. Only a representative sample of 6200 data equivalent to 10 days of each season of the year was used from all the data. An MSE error of 0.078 was obtained. The ANFIS controller performed better than the previous ones under different atmospheric conditions, achieving an efficiency of 99.3%.

In [76] the authors develop a hybrid system, which combines the FL with Hopfield ANN (HNN) of 16 neurons to optimize its membership functions. The model was first simulated in Matlab, using input I and V from the PV panel, and as output generates PWM output with the generated reference load. The algorithm was also tested using a laboratory simulator using a SPACE DS1104 board and an Agilent E4360A 1 kW PV simulator. It is shown that the algorithm can obtain the MPP very similar to the real one, and under changing G and T conditions, it also obtained superior response times against the P&O algorithm. The author considers that it is a good candidate for creating MPPT connected to the public grid.

In [77] present a neuro-fuzzy adaptive control that optimizes the parameters and the member function. The fuzzy function in used to calculate the MPP region and then, the obtained parameters are passed to a P&O algorithm. The performance of the algorithm was tested under changing environmental conditions. Finally, the algorithm overcomes the limitations of using each algorithm separately.

In [78], the authors present an ANFIS for MPPT control in PV systems. This model was simulated in Matlab using the included ANFIS editor. One hundred ninety datasets were used, and a table with 50 membership functions was created from which nine logic rules were generated for MPP calculation. The results show that it can perform in changing weather conditions, and it has a short response period. The controller shows that the MPP increases when G increase and T stays moderate, demonstrating a PV nonlinear behavior.

Ref. [79] develop a neuro-fuzzy InC algorithm with variable step using the same input variables I&V and where the output is a PWM controlling the DC/DC output. It uses 25 rules for training and uses the Mamdani inference system. It has a response time of 15 ms against the 60ms of the InC. The Matlab simulation proves that it is more efficient than its counterpart in different environmental conditions, and it leads to improved power output and reduced energy losses.

In [80], the authors present a hybrid MPPT control algorithm for PV systems, which merges ANFIS with the bee colony optimization algorithm. This type of algorithm, according to the authors, has not been discussed before. The results obtained validate the hypothesis presented. The percentage of effectiveness of 98% was compared to genetic ANFIS and particle ANFIS algorithms. The results were simulated in an environment of changing conditions.

In [81] the authors present an MPPT control algorithm for PV systems based on Adaptative FL (AFL) combined with a radial-based network (RBNN) for better performance. This approach comes in two stages. The first stage takes the InC algorithm and couples it with FL. The second stage used a Boost converter to maximize the optimal V from the first stage. The simulations show that the algorithm is capable of performing MPPT without perturbations. The algorithm was compared against P&O and against a simple radial-based algorithm.

## 5. MPPT Control Using ANN and Hybrid Metaherustic Algorithms

Hybrid intelligent systems (HIS) combine several of the algorithms or techniques shown above, either as part of a problem-solving method or to perform a specific task, to which may be added a third or fourth technique, a metaheuristic or optimization algorithm to improve ANN performance or improve input parameters. Figure 6 shows the general scheme of an MPPT controller using an ANN algorithm combined with GA to optimize the input and also combined with a traditional P&O algorithm to improve its performance. Additional selected characteristics are shown in Table 3.

In [82], the authors develop an EANFIS algorithm combined with PSO for MPPT control for SAPV systems. This system features a Quasi-Z-source inverter (qZSI) and a brushless motor (BLCD). The algorithm was designed using the Matlab toolbox, and it uses two inference rules and G&T as inputs. Different materials such as sand, dust, and ash were used to simulate different environmental conditions. It was compared to already established algorithms such as P&O 80% and ANN MPPT 87%. The final results show a maximum efficiency of 94% for this proposed algorithm.

Ref. [83] presents a hybrid algorithm for MPPT, using ANFIS combining P&O in a first stage and modified firefly optimization (MFA) in a second stage. The proposed method takes as input different levels of G, where it is processed by the ANFIS and yields an optimal output, which is then given to the P&O algorithm to search for the global MPP. The results show that the proposed algorithm identifies and tracks the MPP in an estimated time of 0.2 s and succeeds in doing so for various G patterns.

In [84], the authors propose an MPPT algorithm using ANFIS + PSO to optimize the membership functions and to reduce the error of the least-squares algorithm. The proposed ANFIS uses the Takagi-Sugeno-Kang system with five layers and opts for four rules for the membership functions. The algorithm uses PSO to find the global optimum and avoid becoming stuck, which achieves stability and ensures convergence. The results were simulated in Matlab/Simulink, obtaining better response time than a traditional P&O, and were tested at different I levels.

Research from [85] proposes a neuro-fuzzy control algorithm. The data were trained on a 3-layer ANN with 25 neurons and were optimized with a GA. The algorithm achieved decreased transient time tracking, negligible variation of the steady-state output power, accurate MPP tracking with changes in weather conditions, and converter load. Finally, increasing the efficiency of the MPPT controller.

In [86] authors developed a MPPT algorithm using hybrid ANN + PSO to find the global peak (GP) in parecence of several local peaks (LP). The performance of this algorithm was compared against a classical swarm algorithm. The algorithm was tested in both Matlab and hardware simulations and obtained very similar efficiency results to each other. However, when tested in partial shading conditions it had some differences.

The authors of [87] proposed a method to calculate the GMPP using the sequential Monte-Carlo method (SMC) combined with an InC and refined through an ANN. The ANN is in charge of tracking the changes of V and I, and G and predicts the GMPP. The SMC algorithm is in charge of the nonlinear transition of V when the step-size is variable. The ANN was trained with 500 data samples, and the simulation results show that the algorithm is good at finding the GMPP under partial shading conditions.

The authors of [88] make use of two algorithms to find the MPP. The first one uses the FL combination with GA and PSO, and it was compared against the P&O and InC tractional, obtained good results in low G and high T conditions. The second one opted for an ANN plus GA with 28 nodes in the hidden layer and a gradient descent training algorithm. The algorithm proves in the simulation to be more effective against the traditional P&0 and InC in high G and low T conditions. Given the results, the author proposes combining the two algorithms to squeeze the most out of the PV system. The results show that the GA + ANN + FL combination extracts the most energy.

The work of [89] propose an MPPT algorithm using a hybrid neural + PSO technique. This was tested under partial shading conditions and against the basic PSO algorithm. The ANN takes irradiance from different sensors but these can be faulty which can be a problem for real systems. It proposes the elimination of these sensors by only observing the I-V curve, with the disadvantage that more data and a better trained ANN would be needed.

In [90] this paper uses a hybrid adaptive neuro-fuzzy algorithm with PSO, one of the advantages of this algorithm is that it does not need the use of sensors to calculate the G and T. This method has better PV power tracking capability, shorter RMSE runtime, and free derivation to find the above parameters for proper training under uniform, non-uniform and changing shading conditions.

Ref. [91] the author proposes two algorithms, the first hybrid combined ANN with Bayesian regulation (BR) and the second combining GA + ANN. The results are checked against simple P&O. The BR is used to search for MPPs at any value of T and G in the first case. The data obtained using GA optimization is used to train the ANN in the second case. The results show an MPPT controller efficiency percentage of 87%, a MAPE error percentage of 0.00842099%. The author recommends using more training data to reduce the error further. Simulations were done using Matlab/Simulink ANN toolbox.

## 6. Discussion

One of the most striking details is that of the total number of papers reviewed, only three published the source code of their work and implementation in a public way. Other articles do their checks in MATLAB/SIMULINK; however, there was no way to access these simulations. Matlab/Simulink, and in particular the ANN Toolbox, was the 100% option used by all the papers to perform checks on their algorithms.

Here is an area of particular interest for developing other platforms/languages, such as the python PVLIB library for simulating PV systems, in addition to taking advantage of the extensive AI and ANN library. Another program of interest would be Labview and its built-in capability to simulate electronic circuits.

The creation of a laboratory project to bridge the gap between theory and practice helps students identify and work with many of the features of PV systems [92]. Including the implementation of many of the algorithms. This provides the following:Create a methodological basis for the analysis of the control of PV systems and their architectures;Detail some of the classical and advanced control algorithms that may be used in them;Enable the to implement the acquired knowledge in a real environment.

Furthermore, in future publications, it is proposed to review these algorithms in a real-world hardware implementation, remembering that MPPT controllers are embedded systems (ES), and taking advantage of the capacity of micro-computers or microcontrollers has been increasing in recent years. The implementation of AI algorithms, especially those using ANN, is no longer a serious problem.

Although there are equations and formulas already established to calculate the network error, which is a sought-after parameter in control theory and ANN, only some articles mention this point [58]. Equations such as Mean absolute percentage error (MAPE), Root mean square error (RMSE), Mean bias error (MBE), and Mean absolute error (MAE) help us to know the performance of our algorithm [23].

MPPT performance can be calculated by the MPPY efficiency formula [57]. Other MPPT performance measures by [65] can be applied, such as response time, overshoot, and ripple.

## 7. Conclusions

Although neuro-control is nothing new, it continues to have a powerful impact in recently booming areas such as renewable energy, with a particular emphasis on PV systems. In this article, a systematic review of state-of-the-art papers from the last six years focused solely on ANN for MPPT control in PV systems. Thirty-three papers were shown that included ANN algorithms and their combination with others such as FL and metaheuristic algorithms. The papers came from different bibliographic sources, with a particular emphasis on the quality of the documents, looking for authors who were leaders in the publication of this specific topic.

The controllers using ANN only reported an average efficiency of 98%. They have the advantage of having a very fast convergence time and are easy to implement, and they are very effective in uniform and varying atmospheric conditions. Their robustness comes from how well the network is trained, so large amounts of data will be needed for the network. If the PV system fails for some reason, it can be trained with new data and still perform acceptably. According to [79] there could be complications when using PV panels of different brands combined at the time of training the network.

ANFIS controllers also obtained good results; an average of 97% efficiency was reported in the MTTP controller. It has the advantage of being easy to implement; although it requires previous knowledge of the system, it is a little heavier at memory level and requires more space than one that uses ANN only. It also has the inherent disadvantages of fuzzy logic, where the fuzzy rules and membership rules must be obtained very well.

ANN + metaheuristic algorithms, as is the case of [85] where it is combined with GA. Here, the GA is in charge of optimizing the ANN, reporting decreasing tracking time, negligible output power ripple in a steady-state, accurate output power tracking in a steady-state, accurate MPP tracking with varying atmospheric conditions. The average efficiency of 96% was reported under partial shading and varying climate weather conditions.

Some of these algorithms are proposed to detect errors or failures in PV systems. PV systems are vulnerable to failures or malfunctions. Increasing their lifetime is also as important as improving their efficiency. Preventive actions must be taken to ensure not only serviceability but also safety. Filling the system with sensors to detect faults and regular operation is too costly and redundant [93]. Using intelligent MPPT algorithms can be a double-edged sword of course, it increases the system efficiency, but it also masks different symptoms of faults. Works that combine ANN [94,95] and even FL [96] are already beginning to be explored.

The expectation for this review is that it will be helpful for programmers and developers working in control and renewable energy, especially for those seeking to develop digital MPPT devices in hardware for PV systems. The future direction of research is that neural networks and hybrid algorithms will play an important role in renewable energy development.

The future works in the application of these techniques could help in the design and development of a hybrid neural algorithm for PV systems. Possible implementation in Python and C++. Furthermore, in the creation of a web repository containing the simulation of the PV system and the MPPT controller in MATLAB/Simulink and other languages, and of the most common algorithms such as P&O, InC, and their improvements, as well as those that make use of FL, ANN. It could be possible to design and implementation in the hardware of a reprogrammable MPPT controller, with an open license, to test the newly designed algorithms. One proposal could be the use of 32-bit ARM microcontrollers for this function. Furthermore, try the feasibility of using small board computers (SBC) such as the RaspberryPi to create an MPPT. Incorporate battery management system (BMS) algorithms in MPPT controllers. Finally, to study in-depth the operation of metaheuristic algorithms and their application to ANN and control systems such as :Ant lions algorithmBAT algorithmBlack Widow AlgorithmDragonfly AlgorithmGrasshopper Optimization AlgorithmMoth Flame Optimization AlgorithmMultiversal OptimizationSalp Swarm Optimization Algorithm

## Figures and Tables

**Figure 1 micromachines-12-01260-f001:**
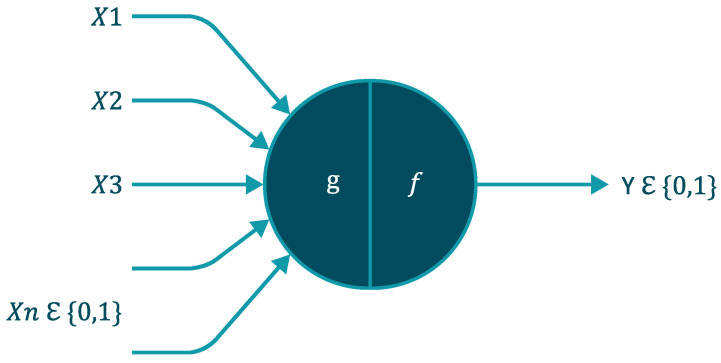
Model of an artificial neuron.

**Figure 2 micromachines-12-01260-f002:**
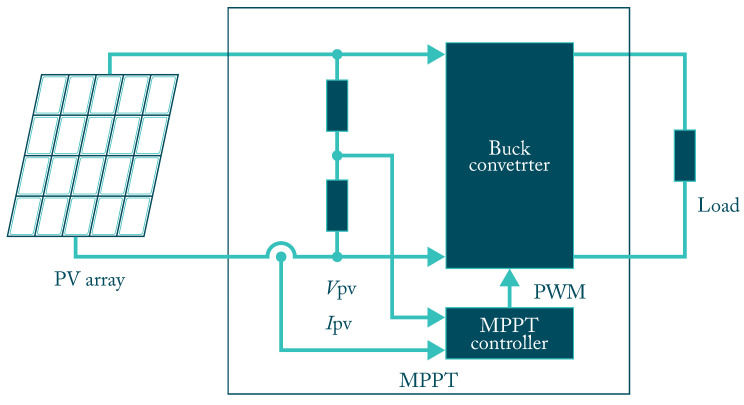
Stand alone SAPV system with MPPT inverter.

**Figure 3 micromachines-12-01260-f003:**
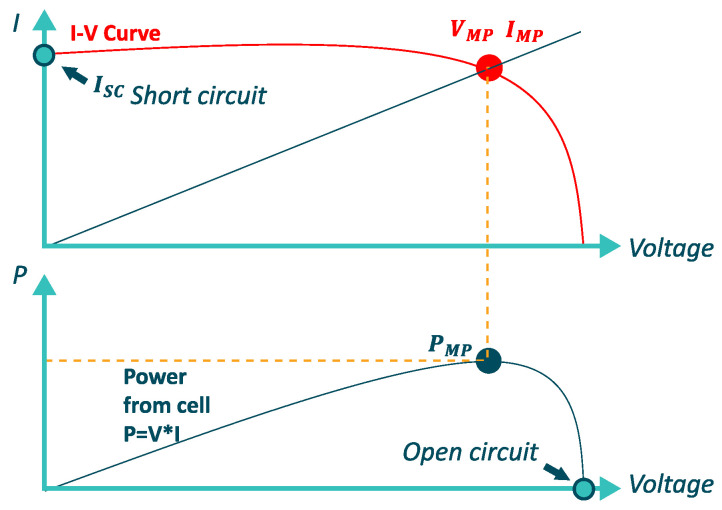
Charateristic I-V curve of a PV module.

**Figure 4 micromachines-12-01260-f004:**
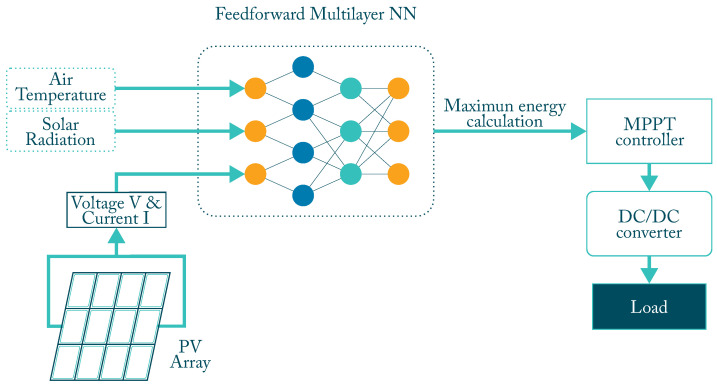
Block diagram of an ANN MPPT controller.

**Figure 5 micromachines-12-01260-f005:**
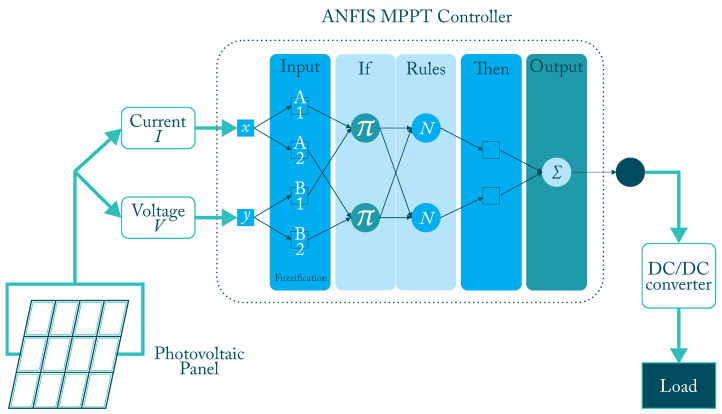
ANFIS controller diagram block with 2 inputs.

**Figure 6 micromachines-12-01260-f006:**
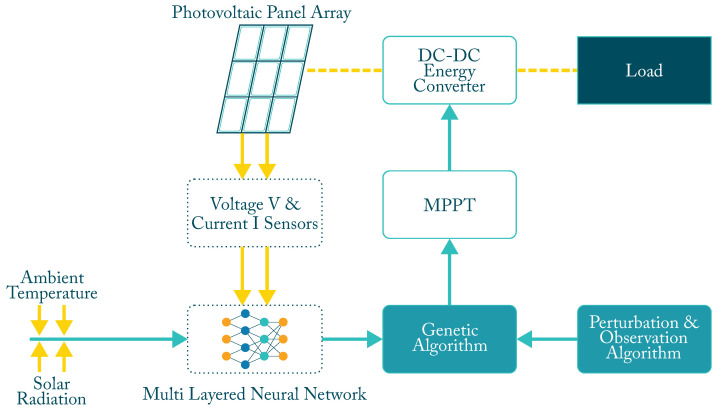
Block diagram of an hybrid MPPT controller.

**Table 1 micromachines-12-01260-t001:** Feature review for MPPT controller agorithms using ANN.

References	Network Architecture	Efficiency %	Conditions	Error	Input Variables
[57]	multilayer feedforward	99.6%	variable	–	*G* and *T*
[58]	backpropagation momentum	98.47%	variable	–	*V* and *I*
[59]	multilayer feedforward	–	uniform	–	*V*, *I*, *G*, *T*
			and variable		
[60]	Rprop-NN	–	variable	–	*G* and *T*
[61]	Recurrent exogenous (NARX)	–	partial shading	–	*V* and *T*
[62]	single-layer	–	uniform	0.0612	*G* and *T*
[63]	RHONN	93%	variable	0.3131	–
[64]	feedforward three layers	98.77%	uniform	–	*V* and *I*
			and partial shading		
[65]	multilayerd feedforward	–	variable	0.094115	power derivate (dP)
					and voltage derivate (dV)
[66]	feedforward	97%	variable	2.1961	*G* and *T*
[67]	NARX	–	variable	0.0159	*G* and *T*

**Table 2 micromachines-12-01260-t002:** Feature review for MPPT controller agorithms using ANN + FL (ANFIS).

References	Number of Rules	Type	Efficiency% of MPPT Controller	Fuzzy Inference System	Network Architecture
[71]	9	ANFIS	80%	Sugeno Model	–
[72]	6	ANFIS	–	Sugeno Model	feedforward
[73]	42	ANFIS	–	Sugeno Model	–
[74]	49	ANFIS	99.88%	Sugeno Model	–
[75]	15	ANFIS	99.3%	Sugeno Model	backpropagation
[76]	19	ANFIS	–	Sugeno Model	Hopfield NN
[77]	10	ANFIS	85%	Sugeno Model	–
[78]	50	ANFIS	–	Sugeno Model	–
[79]	25	Neuro Fuzzy	–	Sugeno Model	Variable Step Size
[80]	25	ANFIS-ABC	98.39%	Sugeno Model	–
[81]	2	Fuzzy Adaptive	99.21%	Mamdani	RBF-NN

**Table 3 micromachines-12-01260-t003:** Feature review for MPPT controller agorithms using hybrid (HIS) algorithms NN + metaheuristic.

References	Type	Efficiency %	Metaheuristic Algorithm	Network Architecture	Inference System
[82]	EANFIS+PSO	94%	Particle Swarm	–	Sugeno Model
[83]	FA-ANFIS-P&O	–	Modified Firefly Algorithm	–	Takagi-Sugeno
[84]	ANFIS–PSO	97%	Particle Swarm	RLSE	Takagi-Sugeno-Kang
[85]	Hybrid (Fuzzy+NN+GA)	–	Genetic	MLP	–
[86]	Hybrid (NN+PSO)	92.7–99.7%	Particle Swarm	Backpropagation	–
[87]	Hybrid (NN+SMC)	96.2%	Secuential Monte-Carlo	MLFF	–
[88]	Hybrid (ANN+GA)	–	Genetic	MLP	–
[89]	Hybrid (ANN+PSO)	99.89%	Particle Swarm	FFBP	–
[90]	ANFIS–PSO	98.35%	Particle Swarm	MFNN	max–min Mamdani
[91]	Hybrid (ANN+GA+BR)	85%	Genetic	Bayesian Regulation	–

## Data Availability

The data presented in this study are available on request from the corresponding author.

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
