# Peer review of "Artificial Neural Networks in MPPT Algorithms for Optimization of Photovoltaic Power Systems: A Review"

_micromachines, 2021, doi:10.3390/mi12101260_

Round 1
Reviewer 1 Report
A very interesting summary of MPPT algorithms is discussed in this paper. I have several remarks to point,
1) Please discuss the modelling of the ANN networks in further detail. Perhaps you can add some examples of how to calculate the neurons weighting, hidden layers, etc.
2) Please mention that other MPPT algorithms are now being used to fix other mismatching conditions in PV systems including hotspots [1-3].
[1] Dhimish, M. (2019). Assessing MPPT techniques on hot-spotted and partially shaded photovoltaic modules: Comprehensive review based on experimental data. IEEE Transactions on Electron Devices, 66(3), 1132-1144.
[2] Gosumbonggot, J., & Fujita, G. (2019). Global maximum power point tracking under shading condition and hotspot detection algorithms for photovoltaic systems. Energies, 12(5), 882.
[3] Dhimish, M. (2019). 70% decrease of hot-spotted photovoltaic modules output power loss using novel MPPT algorithm. IEEE Transactions on Circuits and Systems II: Express Briefs, 66(12), 2027-2031.
3) What are the main differences between MLP ANN network and single layer ANN network? I would also comment here (page 12-14) that some MPPT has the functionality to build PV fault detection and classification such as the recent deployed methods [4-6].
[4] Bakdi, A., Bounoua, W., Guichi, A., & Mekhilef, S. (2021). Real-time fault detection in PV systems under MPPT using PMU and high-frequency multi-sensor data through online PCA-KDE-based multivariate KL divergence. International Journal of Electrical Power & Energy Systems, 125, 106457.
[5] Hussain, M., Dhimish, M., Titarenko, S., & Mather, P. (2020). Artificial neural network based photovoltaic fault detection algorithm integrating two bi-directional input parameters. Renewable Energy, 155, 1272-1292.
[6] Vieira, R. G., Dhimish, M., de Araújo, F. M., & Guerra, M. I. (2020). PV Module Fault Detection Using Combined Artificial Neural Network and Sugeno Fuzzy Logic. Electronics, 9(12), 2150.
4) Please comment on page 14 (discussion section) on the inputs parameters usually required to build an MPPT algorithm, and how the precision of the sensors can influence the accuracy of PV maximum power tracking.
Author Response
A very interesting summary of MPPT algorithms is discussed in this paper. I have several remarks to point,
- Please discuss the modelling of the ANN networks in further detail. Perhaps you can add some examples of how to calculate the neurons weighting, hidden layers, etc.
Ans: Thank you for your comment. We have discussed the modeling of the ANN networks in further detail.
- Please mention that other MPPT algorithms are now being used to fix other mismatching conditions in PV systems including hotspots [1-3].
Ans: Thank you for your comment. We have mentioned that other MPPT algorithms and included the following references:
[1] Dhimish, M. (2019). Assessing MPPT techniques on hot-spotted and partially shaded photovoltaic modules: Comprehensive review based on experimental data. IEEE Transactions on Electron Devices, 66(3), 1132-1144.
[2] Gosumbonggot, J., & Fujita, G. (2019). Global maximum power point tracking under shading condition and hotspot detection algorithms for photovoltaic systems. Energies, 12(5), 882.
[3] Dhimish, M. (2019). 70% decrease of hot-spotted photovoltaic modules output power loss using novel MPPT algorithm. IEEE Transactions on Circuits and Systems II: Express Briefs, 66(12), 2027-2031.
3) What are the main differences between MLP ANN network and single layer ANN network? I would also comment here (page 12-14) that some MPPT has the functionality to build PV fault detection and classification such as the recent deployed methods [4-6].
Ans: Thank you for your comment. We have discussed main differences between MLP ANN network and single layer ANN network by adding the following references:
[4] Bakdi, A., Bounoua, W., Guichi, A., & Mekhilef, S. (2021). Real-time fault detection in PV systems under MPPT using PMU and high-frequency multi-sensor data through online PCA-KDE-based multivariate KL divergence. International Journal of Electrical Power & Energy Systems, 125, 106457.
[5] Hussain, M., Dhimish, M., Titarenko, S., & Mather, P. (2020). Artificial neural network based photovoltaic fault detection algorithm integrating two bi-directional input parameters. Renewable Energy, 155, 1272-1292.
[6] Vieira, R. G., Dhimish, M., de Araújo, F. M., & Guerra, M. I. (2020). PV Module Fault Detection Using Combined Artificial Neural Network and Sugeno Fuzzy Logic. Electronics, 9(12), 2150.
4) Please comment on page 14 (discussion section) on the inputs parameters usually required to build an MPPT algorithm, and how the precision of the sensors can influence the accuracy of PV maximum power tracking.
Ans: Thank you for your comment. We have commented on the papers above by adding them to the literature.
Reviewer 2 Report
This is avery well written review that discusses all recent advances in the field of MPPT Algorithms for Optimization of Photovoltaic Power Systems. The authors have organized very well the content provideing very conscise figures which present the ideas. The paper is of interest for peaple working with PVs and thus i recmmnd it for publication in its presnt form.
Author Response
This is avery well written review that discusses all recent advances in the field of MPPT Algorithms for Optimization of Photovoltaic Power Systems. The authors have organized very well the content provideing very conscise figures which present the ideas. The paper is of interest for peaple working with PVs and thus i recmmnd it for publication in its presnt form.
Ans: Thank you for your comment. We improved certain sections according to the other reviewer's comments.